# A Family with Meester–Loeys Syndrome Caused by a Novel Missense Variant in the *BGN* Gene

**DOI:** 10.3390/ijms262412044

**Published:** 2025-12-15

**Authors:** José A. Riancho, Ana I. Vega, Alvaro del Real, Carolina Sañudo, José L. Pérez-Castrillón, Raquel García-López, Nuria Puente, J. Francisco Nistal, José L. Fernández-Luna

**Affiliations:** 1Servicio de Medicina Interna, Hospital U.M. Valdecilla, 39008 Santander, Spain; 2Departamento de Medicina y Psiquiatría, Universidad de Cantabria, 39008 Santander, Spain; 3Instituto de Investigación Marqués de Valdecilla (IDIVAL), 39011 Santander, Spain; 4Centro de Investigación Biomédica en Red de Enfermedades Raras (CIBERER), 46010 Madrid, Spain; 5Unidad de Genética, Hospital U.M. Valdecilla, 39008 Santander, Spain; 6Servicio de Medicina Interna, Hospital Río Hortega, 47012 Valladolid, Spain; 7Departamento de Medicina, Universidad de Valladolid, 47005 Valladolid, Spain; 8Departamento de Fisiología y Farmacología, Universidad de Cantabria, 39011 Santander, Spain; 9Servicio de Cirugía Cardiovascular, Hospital U.M. Valdecilla, 39008 Santander, Spain; 10Departamento de Ciencias Médicas y Quirúrgicas, Universidad de Cantabria, 39011 Santander, Spain; 11Centro de Investigación Biomédica en Red Cardiovascular (CIBER-CV), Instituto de Salud Carlos III, 28029 Madrid, Spain

**Keywords:** connective tissue disorder, aortic aneurysm, biglycan, joint hypermobility, TGF-β

## Abstract

Meester–Loeys syndrome (MLS) is an X-linked connective tissue disorder caused by pathogenic *BGN* variants. We describe a family carrying a novel missense variant. The index male, initially diagnosed with Ehlers–Danlos syndrome, had joint hypermobility, multiple visceral artery aneurysms, and recurrent musculoskeletal problems. A brother of the proband had an aortic root aneurysm. Female carriers had no or only minor manifestations. Studies of the aortic wall were consistent with a dysregulation of the TGF-β/SMAD pathway and assays with reporter vectors revealed reduced canonical Wnt and TGF-β activity in cell lines expressing mutant biglycan. However, patients’ dermal fibroblasts did not show consistent differences in the nuclear abundance of β-catenin or p-SMAD2/3 compared to cells from controls. This 3-generation family expands the genetic and phenotypic spectrum of MLS and underscores the importance of considering *BGN* testing in hypermobility syndromes to enable early surveillance and targeted management.

## 1. Introduction

Hypermobility disorders are a heterogeneous group of connective tissue syndromes characterized by joint hypermobility, musculoskeletal complications, and sometimes systemic manifestations involving the skin, cardiovascular system, and other organ systems.

Ehlers–Danlos syndrome (EDS) encompasses a group of heritable connective-tissue disorders characterized by varying degrees of joint hypermobility, skin hyperextensibility, and tissue fragility. The condition includes multiple subtypes, each linked to distinct autosomal mutations that impair collagen structure or processing. Vascular EDS (vEDS) is defined by marked arterial fragility—manifesting as aneurysms, dissections, or ruptures—often accompanied by variable joint hypermobility. Most cases result from pathogenic variants in the *COL3A1* gene [1]. However, the genetic basis of hypermobile EDS (hEDS)—the most common EDS subtype, characterized by joint hypermobility and skin abnormalities but typically lacking arterial involvement—remains unknown [2].

Marfan syndrome is an autosomal dominant disorder caused by pathogenic variants in *FBN1* (fibrillin-1) and primarily affects the cardiovascular, ocular, and musculoskeletal systems. Its major clinical features include aortic root aneurysm or dissection, ectopia lentis, tall stature, disproportionately long limbs, arachnodactyly, and scoliosis [3].

Loeys–Dietz syndrome (LDS) is characterized by aggressive, early-onset vascular disease, including aortic and branch vessel aneurysms, arterial tortuosity, and aortic dissections, even with mild aortic dilation. The genetic basis of LDS involves heterozygous pathogenic variants in genes encoding components of the transforming growth factor-β (TGF-β) signalling pathway, such as TGFB1/2, TGF-β receptors or the SMAD family members 2 and 3 (SMAD2/3) [3].

Canonical TGF-β signalling begins when TGF-β ligands bind to a heterotetrameric receptor complex composed of type I and type II receptors. Ligand binding activates the type II receptor, which phosphorylates and activates the type I receptor. The type I receptor then phosphorylates receptor-regulated SMADs (SMAD2/3). These form a heterodimeric complex with SMAD4 that translocates to the nucleus and regulates the expression of target genes. In addition, there are other non-canonical pathways to transmit TGF-β signals that do not involve SMAD proteins [4,5,6].

Extensive evidence supports the involvement of primary or secondary abnormalities in the TGF-β pathway in the pathogenesis of arterial disease in Marfan syndrome and LDS, although the precise molecular mechanisms remain incompletely understood. Fibrillin-1 normally sequesters TGF-β within the extracellular matrix, and loss-of-function mutations in *FBN1* disrupt this regulatory function, leading to increased levels of bioactive TGF-β [7]. In LDS, pathogenic variants in genes encoding TGF-β ligands, receptors, or the downstream mediators SMAD2/3 result in dysregulated signalling. Paradoxically, despite carrying mutations predicted to interfere the pathway, patients with LDS exhibit evidence of enhanced TGF-β activity, including increased SMAD2 phosphorylation. This paradox is thought to reflect a compensatory response, but the underlying molecular processes remain to be fully elucidated [8].

Meester–Loeys syndrome (MLS) is a rare, recently described, X-linked syndromic form of connective tissue disorder. The clinical phenotype shows significant overlap with Marfan syndrome and LDS, but is distinguished by its genetic etiology and inheritance pattern. The syndrome is caused by loss-of-function mutations of the *BGN* gene, an X-linked gene which encodes the small leucine-rich proteoglycan biglycan [9]. In their seminal report, Loeys et al. suggested that TGF-β activity was increased in the arterial wall of two patients with MLS. However, the pathogenesis of the disorder is likely complex and involves multiple biological pathways. In fact, biglycan may interact with other signalling pathways involved in matrix homeostasis, including the Wnt pathway [10]. The binding of Wnt ligands to their receptors initiate canonical and non-canonical signalling. The former prevents β-catenin phosphorylation and degradation, allowing it to accumulate in the cytoplasm. Stabilized β-catenin then translocates to the nucleus, where it interacts with TCF/LEF transcription factors to activate Wnt-responsive genes [11].

The literature on MLS is limited, and its clinical phenotype overlaps substantially with other hypermobility disorders, creating significant diagnostic challenges. Distinguishing MLS from hEDS is particularly important because MLS carries a markedly higher risk of life-threatening vascular complications. Here, we describe a family with a novel *BGN* mutation, further expanding the genetic and clinical spectrum of MLS.

## 2. Results

### 2.1. Clinical Data

The index patient, a 53-year-old male, was referred to our clinic with a diagnosis of EDS. He had a long history of joint hypermobility and vascular problems, including aneurysms of the celiac trunk, and the splenic, hypogastric, renal, and iliac arteries, as well as stenosis of the right renal artery (Figure 1). He had acute thrombosis with spleen infarction that required surgery at the age of 37. There were multiple complications, with abdominal wall infections and dehiscence. He had suffered several episodes of luxation of the ankles and fingers, as well as some tooth fractures while eating. The medical history included a traumatic fracture of the wrist and mesenteric panniculitis. The physical exam revealed joint hypermobility (Beighton score 5), with apparently normal skin colour and elasticity. He had radiological signs of osteoarthritis of the spine and the hip. A DXA scan showed low bone mineral density, with T-scores of −1.0 at the lumbar spine and −2.3 at the femoral neck. There were no abnormalities in MRI studies of the brain arteries and the aorta, except for an ectatic aortic root (38 mm). A cardiac ultrasound did not reveal valve abnormalities. A prior genetic study identified a variant of unknown significance (VUS) in *COL1A2* (Arg708Gln), which led to the suggestion of a diagnosis of EDS of unclear type.

The patient had two brothers. One did not show signs of connective tissue disease. The other one (II.2) was 48 years old when first seen in the clinic. Medical history included sleep apnoea, colonic perforation after diverticulitis and colonoscopy, inguinal hernia, multiple ankle and knee sprains, asthma, and hypoacusis. He exhibited hypermobility of the hands (Beighton score 4), and mild skin hyperelasticity. The skin and the scars looked otherwise normal. He had an aneurysmatic aortic root (48 mm), which required surgery. Cardiac valves were normal. Imaging studies did not show abnormalities of the cranial or abdominal arteries.

The daughter of the index patient was a 20-year-old woman with a history of surgically corrected atrial septal defect in childhood, multiple ankle and wrist sprains, and chronic musculoskeletal pains. The physical exam revealed marked hypermobility (Beighton score 8) and normal skin. Cardiac and abdominal ultrasound and brain MRI revealed no abnormalities.

A cousin of the index patient showed mild joint hypermobility during the physical exam (Beighton score 4), repeated ankle sprains, and hip subluxation. Imaging studies revealed a mild dilatation of the aortic root. Other family members showed no signs of connective tissue disease.

### 2.2. Genetic Analysis

Exome analysis of the index patient revealed a hemizygous missense variant of uncertain significance (VUS) in the *BGN* gene (NM_001711.6:c.284T>C), which alters the biglycan amino acid sequence (p.Leu95Pro). Additional heterozygous variants were identified in *COL1A2* (Collagen Type I Alpha 2 Chain; Arg708Gln), *IPO8* (Importin 8; Arg864Gln), and *YYLAP1* (YY1 Associated Protein 1; Arg92Cys), although these were considered less likely to be related to the clinical phenotype. This *BGN* variant is a novel one, not present in GnomAD or disease databases. Several bioinformatic tools predicted a deleterious effect on protein function (AlphaMissense, Ravel, Varity, SIFT). The Leu95Pro variant is allocated in a beta sheet that conforms a hydrophobic pocket that interacts with TGF-β (Figure 2).

Most of the parallel chains of this beta sheet contain a leucine in the middle. Leu, along with Ile and Val are preferred amino acids in parallel beta sheets [12]. On the contrary, Pro has structural limitations and does not favour the formation of beta sheets, and if present, it is located at the ends, not in the middle of the beta chain. Therefore, it is very likely that the Leu to Pro change destabilizes the beta sheet by affecting protein–protein interactions. Accordingly, two different applications for calculating the free energy change (G) caused by the amino acid variant, SAAFEC-SEQ [13] and ELASPIC [14] (Figure 2).

Segregation analysis in additional family members supported the *BGN* variant as the most likely causal candidate (Figure 3).

### 2.3. Functional Studies

Western blot analysis of the aortic wall from patient II.2, obtained at surgery, revealed reduced levels of p-SMAD2/3 abundance compared to a control individual. Likewise, qPCR analyses showed reduced expression (i.e., >2-fold difference) of a number of TGF-β target genes, such as *COL1A1*, *TGFBR3*, *CTGF*, and *LGR5*, but no difference regarding *POSTN* or *SERPINE* expression. The expression of the Wnt target *AXIN2* was also lower than in a control sample. However, *BGN* expression levels were similar in the patient’s and control samples (Figure 4A1,A2)

In experiments using cell lines, the response of p-SMAD2/3 after TGF-β stimulation was attenuated when the mutant *BGN* vector was transfected instead of the wild-type vector (Figure 4B1). Likewise, co-transfection of the TOPFLash Wnt reporter with *BGN* expression vectors into HEK-293T cells demonstrated significantly reduced reporter activity for the mutant *BGN* vector relative to the wild-type vector (Figure 4B2,B3), consistent with diminished canonical Wnt pathway activity.

On the other hand, we did not observe significant differences in fibroblast proliferation rates or cell survival after serum deprivation when comparing patient-derived and control explants. Likewise, *BGN* expression was similar in fibroblasts from patients and controls (ΔCt 2.4 ± 0.1 vs. 1.8 ± 0.8, not significant). As biglycan abnormalities have been suggested to impact TGF-β and Wnt signalling, we next used confocal immunofluorescence microscopy targeting p-SMAD2/3 and β-catenin, critical mediators of those pathways. Studies of cultured skin fibroblasts revealed no consistent differences in p-SMAD or β-catenin abundance between patients and controls (Figure 5). Similarly, sex-stratified analyses did not reveal clear differences between the two groups. Furthermore, the expression of *LGR5*, a TGF-β target gene, was slightly higher in fibroblasts grown from patients’ skin samples than in those from controls (ΔCt −3.3 ± 0.2 vs. −5.3 ± 0.5, *p* = 0.02). Otherwise, we did not detect significant between-group differences in the expression of TGF-β target genes.

## 3. Discussion

MLS is a recently characterized X-linked connective tissue disorder caused by loss-of-function variants in the *BGN* gene, which encodes the small leucine-rich proteoglycan biglycan. Since the first description, MLS has been recognized as overlapping phenotypically with Marfan syndrome, LDS, and certain EDS subtypes, complicating diagnosis (Table 1).

In this study, we describe a family carrying a novel *BGN* variant (p.Leu95Pro) associated with a phenotype compatible with MLS. The clinical presentation of affected individuals included joint hypermobility, musculoskeletal complications, and, in some cases, significant vascular involvement, such as aortic root and visceral artery aneurysms. Notably, disease severity varied, with male members exhibiting more pronounced manifestations, while females were asymptomatic or showed only mild skeletal symptoms, consistent with the X-linked inheritance pattern of MLS. This report expands the mutational spectrum of *BGN* and underscores the phenotypic variability of the condition.

The family segregation analysis clearly pointed to *BGN* as the driver gene. Additional genetic variants were identified in several family members. Variants in *COL1A2* are known to cause osteogenesis imperfecta and rare EDS subtypes, including the arthrochalasia and cardiac-valvular types [15]. Pathogenic variants in *YY1AP1* are associated with a distinct vascular phenotype resembling fibromuscular dysplasia and are frequently accompanied by skeletal and neurodevelopmental abnormalities [16]. Similarly, biallelic loss-of-function variants in *IPO8* underlie a syndromic form of thoracic aortic aneurysm characterized by motor developmental delay, connective-tissue manifestations, and craniofacial dysmorphism (e.g., frontal bossing, hypertelorism, retrognathia, and ptosis) [17]. Given their phenotypic spectra, those variants are unlikely to account for the clinical features observed in this family. However, a potential modifying effect on the *BGN*-led phenotype cannot be entirely ruled out.

The seminal paper by Meester et al. on the now-called MLS described 5 individuals with loss-of-function mutations of *BGN* among 11 patients with molecularly unexplained thoracic aortic aneurysm and dissection [18]. Consequently, all patients with *BGN* mutations had aortic pathology. However, more recent data suggest a broader and variable multisystem phenotype of MLS [19]. Our findings align with this broader view, as some carriers in this family displayed mild or absent vascular findings, yet retained musculoskeletal features of a connective tissue disorder. The fact that men are mainly affected is consistent with a recessive X-linked inheritance. However, the occurrence of affected females in the present family, as well as in a previous report, suggests that the disorder may exhibit a semi-dominant mode of heredity with reduced penetrance and variable expressivity in females. Nevertheless, given the scarcity of publications on MLS, the true prevalence of abnormalities among women remains uncertain.

Previously reported variants causing MLS were usually frameshift, nonsense, or splice site [19]. However, in this family, the disorder was related to a missense variant, thus expanding the mutational spectrum of the disease.

Biglycan is an essential component of the extracellular matrix. There is both clinical and experimental evidence for an important role in the homeostasis of several tissues, including skin, bone, ligaments, vascular wall, and heart, among others [20,21,22,23,24,25]. However, the involved mechanisms are largely unknown. They may include both direct cell signalling by biglycan and the modulation of factors, such as TGF-β and Wnt [26]. In fact, Meester et al. suggested that aortic wall disease was related to abnormal TGF-β activity [18]. Moreover, impaired Wnt signalling may represent a mechanism underlying reduced bone anabolism in the context of abnormal biglycan activity [10,21,27].

Our data show that the Leu95Pro variant results in significantly reduced canonical Wnt pathway activity in vitro, as evidenced by lower TOPFlash reporter activity in HEK-293T cells. On the other hand, gene expression analysis in the aortic wall suggested reduced levels of several TGF-β target genes and the Wnt target *AXIN2*. However, we found no consistent differences in gene expression or in p-SMAD or β-catenin levels between fibroblasts from patients and those from controls. These contradictory results are not entirely unexpected, given the well-known context-dependent nature of TGF-β signalling and effects [6]. Furthermore, biglycan has a complex and incompletely understood mechanism of action [26,28]. It is widely regarded as a modulator of TGF-β activity. However, the effects of biglycan may vary across tissues and in relation to specific contextual conditions. For example, in experiments with knock-out mice, biglycan deficiency was associated with increased TGF-β activity [24]. On the other hand, biglycan binds TGF-β and the TGF-β type I receptor (ALK5), enhancing ALK5-SMAD2/3 signalling and thus intensifying TGF-β activity in vascular endothelial cells [29]. Therefore, although our results are consistent with the occurrence of complex abnormalities of TGF-β and Wnt signalling, the exact pathogenic mechanisms responsible for the vascular abnormalities in MLS remain unclear.

Our study adds to the limited literature on MLS by documenting a novel missense *BGN* variant and providing genotype–phenotype correlations in a multi-generational family. It also highlights the diagnostic challenges posed by MLS, particularly in differentiating it from EDS if striking vascular disease is absent. In this case, prior consideration of an EDS diagnosis delayed recognition of MLS, which carries a substantially higher vascular risk and, consequently, different surveillance and management implications.

Strengths of this work include the combination of detailed clinical characterization, segregation analysis, and functional assays to support the pathogenicity of the novel variant. The inclusion of mildly affected female carriers and mildly symptomatic relatives provides insight into the variable expressivity and sex-related differences in MLS. Furthermore, our functional data contribute to the growing understanding of biglycan’s role in vascular and connective tissue homeostasis. Given the present and previous observations suggesting that TGF-β signalling is involved in MLS, it may be worth trying angiotensin receptor blockers and/or beta blockers in MLS, for these drugs are usually well-tolerated and have demonstrated benefits in other disorders with TGF-β-related pathogenesis, such as Marfan and Loeys–Dietz syndromes [30,31].

Limitations should also be acknowledged. The limited number of patients precluded a comprehensive analysis of the MLS phenotype. Functional studies were performed in single arterial samples and only a few fibroblastic cell lines. It is worth mentioning that the effects of factors such as biglycan and TGF-β are tissue and context-specific. Hence, cultures of skin-derived fibroblasts may not fully recapitulate in vivo biology, particularly that of vascular tissues. Additionally, the absence of longitudinal imaging data for some relatives limits conclusions about the penetrance and progression of this disease.

In conclusion, this report of a family with MLS emphasizes that this disorder should be considered in the differential diagnosis of joint hypermobility syndromes, particularly when there is any evidence of arterial involvement, even if mild. Early molecular diagnosis is crucial, as it enables targeted cardiovascular surveillance and appropriate counselling for at-risk relatives. Further studies are needed to elucidate the precise molecular mechanisms involved in the skin and vascular abnormalities that characterize this syndrome.

## 4. Materials and Methods

### 4.1. Clinical Procedures

A patient was referred to our clinic with the diagnosis of Ehlers Danlos syndrome. Clinical data of the patient and other family members are described in Results.

The study was approved by the IRB (CEIMc protocol number 2024.125). After patients’ informed consent, a punch skin biopsy was obtained from the two most affected males (II.1 and II.2) and the women with mild symptoms (III.1) under local anaesthesia. Fibroblasts were grown from the biopsies as previously reported [32]. (After confluency, cells from passages 2–5 were used for the experiments. The results were compared with control fibroblasts obtained from healthy subjects with the same protocol. They included two men (65 and 56 years old), and two women (25 and 27 years old).

Patient II.2 exhibited an ascending aortic aneurysm which was considered an indication for surgical resection. At the operation, the whole surgical specimen was harvested, and samples of the aortic wall were immediately preserved in RNAlater^®^ (Invitrogen, Thermo Fisher Scientific, Carlsbad, CA, USA) for gene and protein expression studies. A similar sample of an organ donor subject was used as control.

### 4.2. Genetic Analyses

A whole-exome study of peripheral blood DNA was carried out (Twist Bioscience, South San Francisco, CA, USA). Library sequencing was performed using the NovaSeq 6000 System™ (Illumina, San Diego, CA, USA), and bioinformatics analysis was performed using DRAGEN™ BioIT Platform software (Illumina, version 07.021.572.3.6.3).

Quantitative and qualitative assessment of the extracted DNA sample was performed. Exonic regions and flanking intronic sequences (+/−10 bp) of more than 20,000 genes included in the Twist Bioscience^®^ for Illumina^®^ Exome 2.5 Panel were captured and enriched, followed by massive paired-end sequencing using the Illumina NextSeq1000™ platform. Bioinformatic analysis was carried out, aligning reads to the GRCh38 reference genome and applying specific quality filtering criteria. The average sequencing depth was 102×. This analysis enables the identification of single nucleotide variants (SNVs) and small insertions/deletions (INDELs) of up to 20 nucleotides. Population databases (ExAC 0.3, gnomAD 2.1.1, and 1000 Genomes Phase 3) and predictive algorithms (PolyPhen, SIFT, MutationTaster, Mutation Assesor, LRT, FATHMM, MetaSVM) were used for variant annotation. Identified genetic variants were named according to the Human Genome Variation Society (HGVS) recommendations and classified following the American College of Medical Genetics and Genomics (ACMG) criteria. Segregation analysis was subsequently conducted by performing Sanger sequencing in available family members.

### 4.3. Fibroblast Proliferation

Cell proliferation was studied in the presence of 2–20% FCS and cell survival after serum deprivation was assessed by estimating cell numbers with the MTT assay.

### 4.4. Cloning and Transfection Experiments

*BGN* cDNA (NM_001711) tagged with Myc-DDK and cloned into pCMV expression vector (Origene Technologies, Rockville, MD, USA) was used as a template to introduce c.284T>C change, by using the QuickChange site-directed mutagenesis kit (Agilent Technologies, Santa Cruz, CA, USA).

The MG-63 cell line was used for transfection studies. Cells were transfected with wild-type or mutant constructs by using lipofectamine 2000 (Invitrogen, Carlsbad, CA, USA). After 48 h, the medium was removed and cells were washed with PBS before harvesting. For TGF-β treatment, following PBS washes, cells were incubated overnight in medium containing 0.1% FCS. The medium was then replaced with fresh medium containing 0.1% FCS and a final concentration of 5 ng/mL TGF-β, and cells were incubated for 90 min prior to harvesting.

To monitor Wnt activity, we used the TOP-Flash reporter (initially provided by Randall Moon, University of Washington, Seattle, WA, USA). This is a luciferase reporter of β-catenin-mediated transcriptional activation which contains 8 TCF/LEF binding sites. HEK-293 cells were co-transfected with 500 ng of either the wild-type or the mutant *BGN* vector together with 200 ng of the TOP-Flash reporter plasmid. Luciferase activity was measured using the Luciferase Assay System (Promega, Madison, WI, USA).

### 4.5. Western Blot

We analyzed p-SMAD2/3 as an index of TGF-β activity. Protein extracts from MG-63 cells were separated on 10% polyacrylamide-SDS gels and transferred to nitrocellulose. Blots were incubated with mouse anti-β-catenin antibody 6101538 (BD Biosciences, Wokingham, UK), rabbit anti-phosphoSmad2/3 antibody AP0548, and rabbit anti-BGN antibody A5770 (both from ABclonal Technology, Woburn, MA, USA). This was followed by incubation with secondary antibodies conjugated to horseradish peroxidase (Santa Cruz Biotechnology, Dallas, TX, USA).

### 4.6. Immunofluorescence Microscopy

Fibroblasts were cultured on microscope glass coverslips until reaching 70–80% confluence, fixed with 3.7% paraformaldehyde and permeabilized with 0.25% Triton X-100 for 15 min. After several washes with PBS and 0.05% PBS-Tween, coverslips were incubated overnight at 4 °C with Phospho-Smad2-S465/467 + Smad3-S423/425 Rabbit pAb (ABclonal) or Anti-β-Catenin antibody (Abcam, Cambridge, UK). Following washes, cells were incubated for 45 min at room temperature with Goat anti-Rabbit IgG (H + L) Cross-Adsorbed Secondary Antibody, conjugated to Alexa Fluor 488 (Invitrogen/ThermoFisher, Waltham, MA, USA). Nuclei were counterstained with DAPI (Merck, Darmstadt, Germany), and actin filaments were visualized using Rhodamine Phalloidin (Invitrogen). Regions of interest and fluorescence intensity were analyzed with Image J software v1.52. (available at https://imagej.net/ij/)

### 4.7. Gene Expression

RNA was extracted with Trizol^r^, and reverse-transcribed with Takara kit. Gene expression was then analyzed by real-time qPCR using Taqman assays. The results were expressed as the relative amount in comparison with the housekeeping genes by computing the ΔCt, as the difference between the average threshold cycle (Ct) of the housekeeping genes (TBP, GAPDH, RPL13A) and the Ct of the target gene.

### 4.8. Artificial Intelligence Tools

Artificial intelligence assistance was limited to linguistic editing and rephrasing for clarity. No AI tools were used for data analysis, interpretation, or generation of figures or other original scientific content. The authors take full responsibility for all aspects of the manuscript.

## Figures and Tables

**Figure 1 ijms-26-12044-f001:**
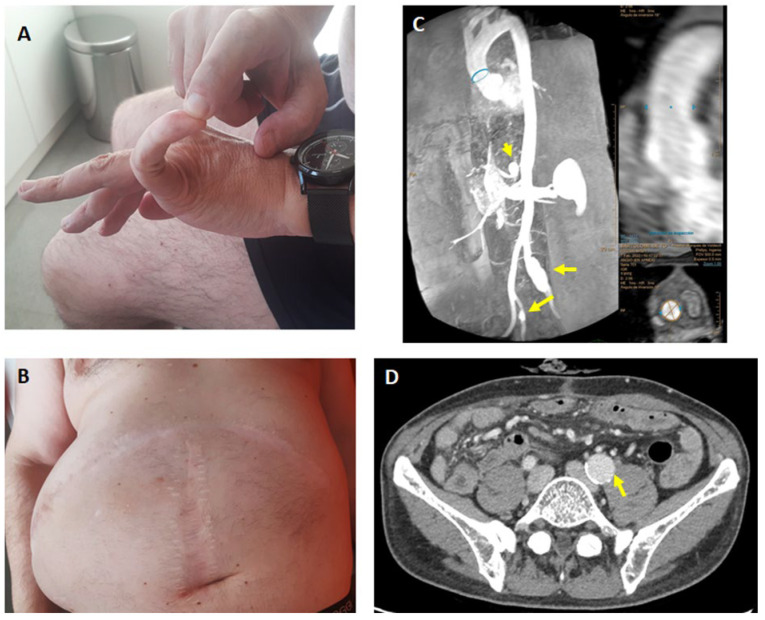
Index patient. (**A**) Finger hypermobility. (**B**) Dehiscent scars and large eventration. (**C**) Angio-MR showing moderate aortic dilatation and aneurysms of the celiac trunk and iliac arteries (arrows). (**D**) CT scan showing an aneurysm of the left iliac artery (arrow).

**Figure 2 ijms-26-12044-f002:**
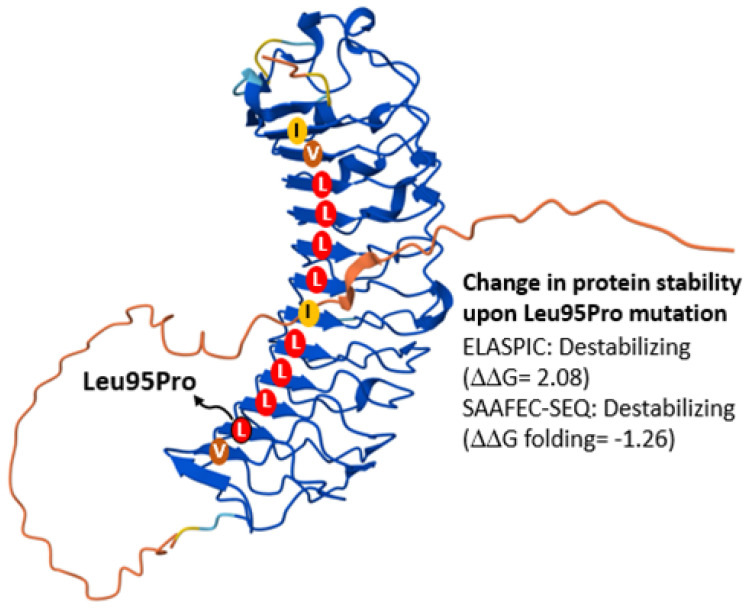
Schematic representation of a 3D model of biglycan showing the hydrophobic pocket that interacts with TGF-β. Values from two different applications for calculating the free energy change (ΔΔG) caused by the amino acid variant, Leu95Pro, are also shown. L, leucine; V, valine; I, isoleucine.

**Figure 3 ijms-26-12044-f003:**
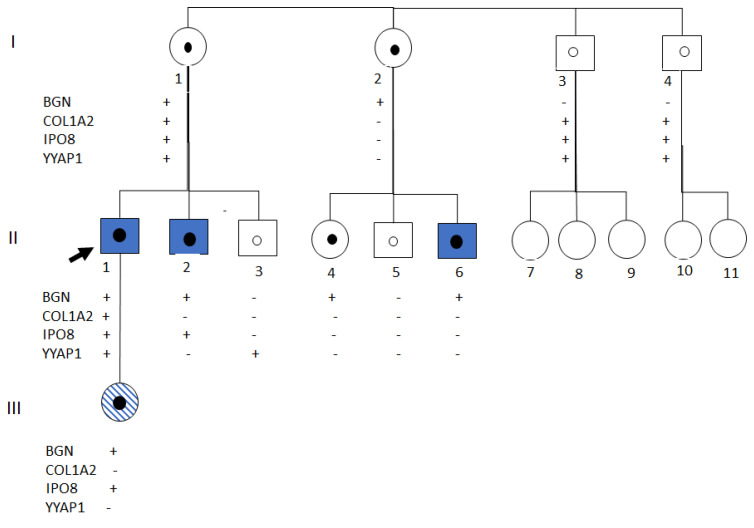
Family pedigree. +, Gene variant present. -, Gene variant absent.

**Figure 4 ijms-26-12044-f004:**
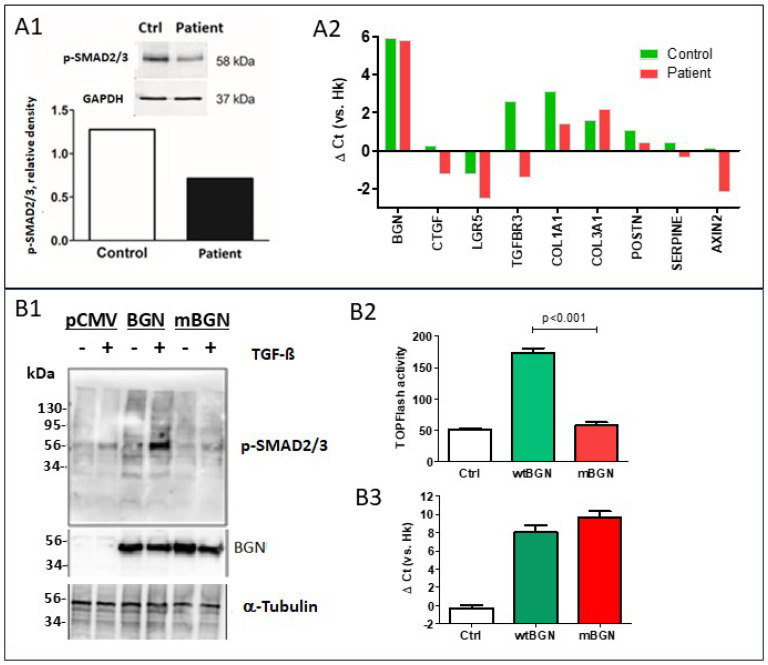
**Functional studies. Aortic wall.** (**A1**) Western blot with p-SMAD staining in patient and control samples. (**A2**) Gene expression in patient and control samples, expressed as the ΔCt (threshold cycle (Ct) of the housekeeping genes minus Ct of the target gene). **Transfection experiments**. (**B1**) Expression of p-SMAD2/3 in MG-63 cells transfected with wild-type or mutant *BGN* (mBGN) constructs in the presence or in the absence of TGF-β was determined by Western blot analysis. Expression of exogenous *BGN* and the α-tubulin control is also shown. (**B2**) Luciferase activity after co-transfection of the TOPFlash reporter with *BGN* expression vectors (wild-type or mutant). (**B3**) *BGN* expression. In both cases, Ctrl experiments included an empty vector. Hk, housekeeping genes.

**Figure 5 ijms-26-12044-f005:**
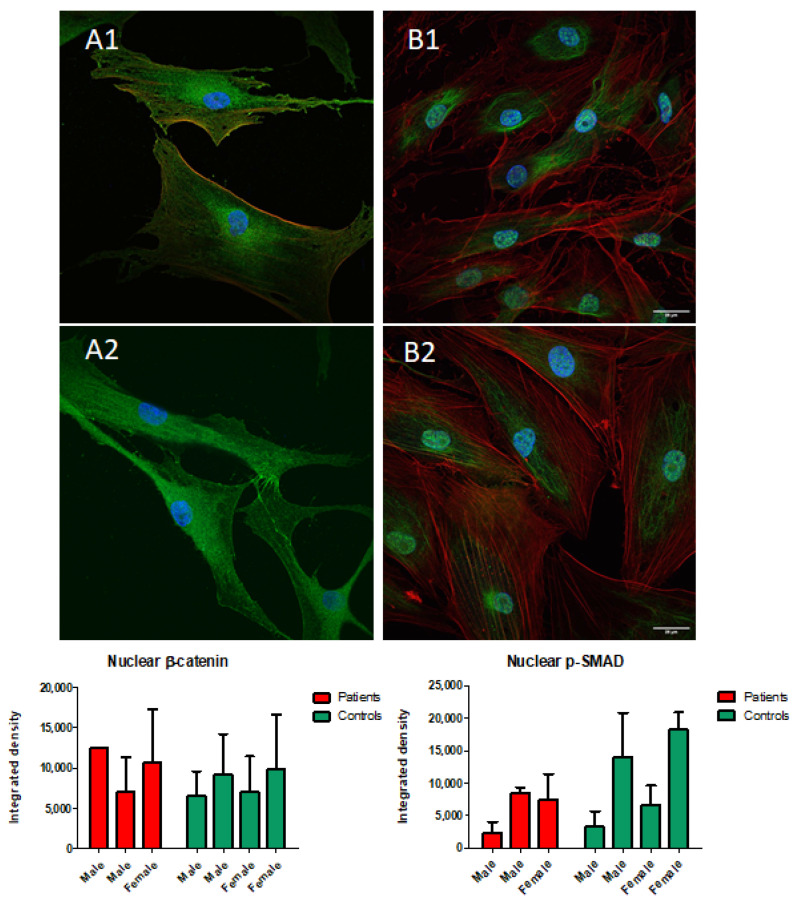
Immunofluorescence of skin-derived fibroblast cultures. β-catenin staining (green) in fibroblasts from one patient (**A1**) and one control (**A2**). p-SMAD2/3 staining (green) in fibroblasts from one patient (**B1**) and one control (**B2**). Nuclei were stained blue with DAPI, and the red signal corresponds to actin filaments. The lower panels show the Individual fluorescence intensity measurements for β-catenin (left) and p-SMAD2/3 (right) in 3 patients and 4 controls. Scale: The bar at the lower right corner corresponds to 25 μm (all images are at the same scale).

**Table 1 ijms-26-12044-t001:** Main features of various connective tissue heritable syndromes.

Syndrome	Meester–Loeys	Loeys–Dietz	Vascular Ehlers-Danlos	Marfan
**Prevalence**	Very rare	Rare	1:50,000–150,000	1:3000–1:5000
**Causal Gene(s)**	*BGN*	*TGFBR1*, *TGFBR2*, *SMAD2*, *SMAD3*, *TGFB2*, *TGFB3*	*COL3A1*	*FBN1*
**Heredity**	X-linked	Autosomal dominant	Autosomal dominant	Autosomal dominant
**Organs** **Affected**	Cardiovascular, skeletal, craniofacial, cutaneous, neurological	Cardiovascular, skeletal, craniofacial, cutaneous	Vascular, cutaneous, gastrointestinal	Cardiovascular, ocular, skeletal
**Main Manifestations**	Aortic/arterial aneurysm and dissection Joint hypermobility	Aggressive aortic/arterial aneurysms and dissections Arterial tortuosity Hypertelorism, cleft palate/bifid uvula Joint hypermobility	Arterial rupture Organ rupture (colon, uterus, lung) Thin skin, easy bruising Mild joint hypermobility Distinctive facial appearance	Aortic root aneurysm/dissection Ectopia lentis Tall stature, long limbs, scoliosis
**Prognosis**	Severe in males, milder in females	Poor	Poor	Variable, improved with treatment

## Data Availability

The original contributions presented in this study are included in the article. Further inquiries can be directed to the corresponding author.

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
