# Peer review of "A Family with Meester–Loeys Syndrome Caused by a Novel Missense Variant in the BGN Gene"

_ijms, 2025, doi:10.3390/ijms262412044_

Round 1

Reviewer 1 Report

Comments and Suggestions for Authors

Family segregation of connective tissue disorder and the novel BGN variants are of interest to readers.  To confirm the gene-disease association, conducting BGN functional expression followed by western blotting and immunohistology is important research topic to identify robust molecular marker specific for Meester- Loeys Syndrome. This syndrome shares overlapping phenotype with other disorders involving vascular and connective tissue pathogenesis such as Loeys-Dietz Syndrome (LDS) and Ehler-Danlos Syndrome (EDS). 

The result section shared the family pedigree demonstrating the segregation of variation within genes BGN, COL1A2, IPO8, YYAP1. Functional analysis involved BGN, pSMAD expression by western blotting and the molecular cross talk with the Wnt was demonstrated through Beta-Catenin, pSMAD quantitation using immunofluorescence technique. Although the data is preliminary, but this is of significant interest and warrant in-depth investigation to validate these findings.

  • In the introduction section of the manuscript, there is no sufficient information regarding the genes, their products, in interaction with molecule, which is researched, studied and elaborated in the result and the subsequent sections of the draft. Without the context, Readers may find it confusing to understand the study and interpret its implication. The suggestion is to present the genes, expression factors and their cross talk leading to the disease in the introduction section, preferably as a figure or flowchart would help readers better understand the topic.
  • MLS, LDS, EDS disorders are outlined in the draft mentioning that these disorders are subject to be misdiagnosed. For the better understanding, representing their overlapping and the distinguished phenotype, and also the molecular involvement, in the tabular form would be effective way to interpret.
  • The result section does not completely validate the hypothesis formulated – that Wnt pathway and TGF signal the upregulation of ECM leading to vascular and connective tissue pathogenesis- as the cellular expression of beta catenin exhibit no significant difference between the patient and the control.

Although the molecular investigation doesn’t fully validate the proposed mechanism of MLS. However, the family segregation pattern and the varied disease severity among the male members of family is in consistence with the X linked inheritance. Nevertheless, the role of BGN in TGF Wnt signaling, resulting in impaired ECM pathways and manifesting as connective tissue and vascular pathogenesis, requires additional data and investigation to confirm.

Author Response

Reviewer 1

Family segregation of connective tissue disorder and the novel BGN variants are of interest to readers.  To confirm the gene-disease association, conducting BGN functional expression followed by western blotting and immunohistology is important research topic to identify robust molecular marker specific for Meester- Loeys Syndrome. This syndrome shares overlapping phenotype with other disorders involving vascular and connective tissue pathogenesis such as Loeys-Dietz Syndrome (LDS) and Ehler-Danlos Syndrome (EDS). 

The result section shared the family pedigree demonstrating the segregation of variation within genes BGN, COL1A2, IPO8, YYAP1. Functional analysis involved BGN, pSMAD expression by western blotting and the molecular cross talk with the Wnt was demonstrated through Beta-Catenin, pSMAD quantitation using immunofluorescence technique. Although the data is preliminary, but this is of significant interest and warrant in-depth investigation to validate these findings.

  • In the introduction section of the manuscript, there is no sufficient information regarding the genes, their products, in interaction with molecule, which is researched, studied and elaborated in the result and the subsequent sections of the draft. Without the context, Readers may find it confusing to understand the study and interpret its implication. The suggestion is to present the genes, expression factors and their cross talk leading to the disease in the introduction section, preferably as a figure or flowchart would help readers better understand the topicR. Thanks for the comment. We have expanded Introduction to briefly describe the TGF-b and Wnt signaling pathways, and specifically the implication of p-SMADs and b-catenin (lines 61-7 and 86-91).

  • MLS, LDS, EDS disorders are outlined in the draft mentioning that these disorders are subject to be misdiagnosed. For the better understanding, representing their overlapping and the distinguished phenotype, and also the molecular involvement, in the tabular form would be effective way to interpretR. Following your suggestion, we have somewhat expanded the description of these disorders and we now include a table summarizing their main features.

  • The result section does not completely validate the hypothesis formulated – that Wnt pathway and TGF signal the upregulation of ECM leading to vascular and connective tissue pathogenesis- as the cellular expression of beta catenin exhibit no significant difference between the patient and the control.

Although the molecular investigation doesn’t fully validate the proposed mechanism of MLS. However, the family segregation pattern and the varied disease severity among the male members of family is in consistence with the X linked inheritance. Nevertheless, the role of BGN in TGF Wnt signaling, resulting in impaired ECM pathways and manifesting as connective tissue and vascular pathogenesis, requires additional data and investigation to confirm..

R. We comment in Discussion on the discrepancies in our various studies, as well as the need for further studies to elucidate the precise molecular mechanisms involved in the pathogenesis of these disorders. This is also stressed in the last sentence of the Discussion (lines 298-308 and 337-8).

Reviewer 2 Report

Comments and Suggestions for Authors
  1. The final sentences of the Introduction (lines 63–69) read as a description of what an introduction should contain rather than as part of the manuscript, and should be removed.
  2. Providing a brief introduction to the other three genes harboring variants—especially IPO8—would strengthen interpretation. This is particularly important because the index patient’s daughter, despite being female, exhibits mild symptoms with variants found in both BGN and IPO8; individual II.4 carries only the BGN variant and is asymptomatic; and I.1 carries all variants but also shows no symptoms. These patterns suggest that the IPO8 variant may be required for phenotype manifestation in females.
  3. When BGN is first mentioned in the Results (line 114), the full name Biglycan should be included to improve readability.
  4. The manuscript does not specify which tissue was used for whole-exome sequencing, which is important for assessing the possibility of somatic mosaicism.
  5. The sequencing depth of the whole-exome data should be reported.
  6. More detail is needed regarding the whole-exome workflow, including the kits and library preparation methods used, rather than stating only that Twist Bioscience performed the sequencing. This information is necessary to assess gene coverage.
  7. “pSMAD2/3 is used as an indicator of TGF-β activity” is currently placed in the Methods and should be introduced in the Results or Introduction. pSMAD2/3 is otherwise not explained before appearing in the assays. The rationale of using pSMAD2/3 as an indicator of TGF-beta activity is missing.
  8. Notation for pSMAD2/3 is inconsistent throughout the manuscript (pSMAD-2/3, pSMAD2/3, p-SMAD2/3). The same issue appears with TOPFLASH (TOP-FLASH, TOPFlash, TOP-Flash, TOP_FLAS). A consistent naming convention is needed.
  9. The manuscript does not show BGN expression levels in control versus patient samples in Figure 4A, which limits the interpretation of downstream phenotypes.
  10. The y-axis label in Figure 4A2 includes “HK,” but the term is never defined.
  11. Lines 148 and 154 both describe TOPFLASH assays but refer to the same figure, even though Figure 4 contains only one set of TOPFLASH data. Clarification is needed on whether these lines refer to different experiments or duplicates of the same one.
  12. The endogenous genetic and expression status of BGN in MG63 and HEK293T cells should be clarified. Additional explanation is needed for why MG63 cells were chosen for the pSMAD2/3 experiment, while HEK293T cells were used for the TOPFLASH assay.
  13. Figure 4B1 should include an internal loading control, such as GAPDH, to validate protein quantification.
  14. Figure 4B2 should show BGN expression levels, as TOPFLASH reporter activity can be influenced by differences in transfection efficiency across samples.
  15. A brief explanation of the biological relevance and mechanism of the TOPFLASH assay would improve readability for nonspecialist readers.
  16. The meaning of the red signal in Figure 5B is not explained and should be clarified.
  17. The manuscript does not address the rationale for assessing β-catenin abundance in Figure 5, which should be justified.
  18. Figure 5 and the Abstract refer to “nuclear β-catenin” and “nuclear p-SMAD,” but these proteins are typically cytoplasmic under many conditions, and they also seem to be in the cytoplasm shown in the immunofluorescence results in Figure 5. The terminology and interpretation should be clarified.
  19. Quantification by immunofluorescence alone is insufficiently rigorous for the claims presented in Figure 5. Western blot or another quantitative method should be included to support these conclusions.
  20. An alternative explanation for the similar levels of pSMAD observed in patient and control samples (Figure 5) could be differences in BGN expression, which should be assessed and discussed.

Author Response

Reviewer 2

The final sentences of the Introduction (lines 63–69) read as a description of what an introduction should contain rather than as part of the manuscript, and should be removed

Thanks for calling our attention to this mistake. It has been corrected.

Providing a brief introduction to the other three genes harboring variants—especially IPO8—would strengthen interpretation. This is particularly important because the index patient’s daughter, despite being female, exhibits mild symptoms with variants found in both BGN and IPO8; individual II.4 carries only the BGN variant and is asymptomatic; and I.1 carries all variants but also shows no symptoms. These patterns suggest that the IPO8 variant may be required for phenotype manifestation in females.

A new paragraph about those variants has been included in the Discussion (lines 250-61)

When BGN is first mentioned in the Results (line 114), the full name Biglycan should be included to improve readability.

We have done so (line 145).

The manuscript does not specify which tissue was used for whole-exome sequencing, which is important for assessing the possibility of somatic mosaicism.

We now state that peripheral blood DNA was used for sequencing (See Methods, line 370))

The sequencing depth of the whole-exome data should be reported.

We now report the average depth was 102x (see Methods)

More detail is needed regarding the whole-exome workflow, including the kits and library preparation methods used, rather than stating only that Twist Bioscience performed the sequencing. This information is necessary to assess gene coverage.

We now provide a more extensive description of the sequencing/analysis pipeline (365-76).

“pSMAD2/3 is used as an indicator of TGF-β activity” is currently placed in the Methods and should be introduced in the Results or Introduction. pSMAD2/3 is otherwise not explained before appearing in the assays. The rationale of using pSMAD2/3 as an indicator of TGF-beta activity is missing.

We now briefly describe the involvement of SMADs in TGF-b signalling.in Introduction (lines 61-7)

Notation for pSMAD2/3 is inconsistent throughout the manuscript (pSMAD-2/3, pSMAD2/3, p-SMAD2/3). The same issue appears with TOPFLASH (TOP-FLASH, TOPFlash, TOP-Flash, TOP_FLAS). A consistent naming convention is needed.

Thanks for the comment. We have modified some abbreviations for consistency throughout the text and figures.

The manuscript does not show BGN expression levels in control versus patient samples in Figure 4A, which limits the interpretation of downstream phenotypes.

We have added BGN expression levels in the aorta (figure 4) and in fibroblast cultures (text). There were no differences between patients and controls (line 191).

The y-axis label in Figure 4A2 includes “HK,” but the term is never defined.

We now explain that HK refers to housekeeping in figure legend.

Lines 148 and 154 both describe TOPFLASH assays but refer to the same figure, even though Figure 4 contains only one set of TOPFLASH data. Clarification is needed on whether these lines refer to different experiments or duplicates of the same one.

Thanks for pointing to this mistake. We have modified the text to avoid confusion.

The endogenous genetic and expression status of BGN in MG63 and HEK293T cells should be clarified. Additional explanation is needed for why MG63 cells were chosen for the pSMAD2/3 experiment, while HEK293T cells were used for the TOPFLASH assay.

HEK293T cells are much easier to transfect than MG63, particularly when multiple vectors are transfected together, thus we preferred HEK293 for experiments using TOPFlash+BGN vectors. Nevertheless, preliminary data in MG63 were in line with the results found in HEK293T (not shown).

BGN expression levels in cell lines was equivalent to that of cells transfected with an empty vector (which is shown in the figure as the control bar (DeltaCt with respect to housekeeping was about -1 in MG63 and -5 in HEK293T)

Figure 4B1 should include an internal loading control, such as GAPDH, to validate protein quantification.

We now include tubulin control

Figure 4B2 should show BGN expression levels, as TOPFLASH reporter activity can be influenced by differences in transfection efficiency across samples.

We now include BGN expression

A brief explanation of the biological relevance and mechanism of the TOPFLASH assay would improve readability for nonspecialist readers.

We now explain that it is a Wnt/betacatenin reporter (Results and Methods)(lines 185 and 397-401)

The meaning of the red signal in Figure 5B is not explained and should be clarified.

This is now explained in Methods (line 420) and the figure legend.

The manuscript does not address the rationale for assessing β-catenin abundance in Figure 5, which should be justified.

We mention the relationship between BGN and Wnt in Introduction (lines 86-91) and Discussion.

Figure 5 and the Abstract refer to “nuclear β-catenin” and “nuclear p-SMAD,” but these proteins are typically cytoplasmic under many conditions, and they also seem to be in the cytoplasm shown in the immunofluorescence results in Figure 5. The terminology and interpretation should be clarified.

Thanks for the comment. As seen in the figures those factors can be found in the cytoplasm. However they exert their gene-regulating effects in the nucleus. Therefore, the nuclear levels are usually considered more relevant.

Quantification by immunofluorescence alone is insufficiently rigorous for the claims presented in Figure 5. Western blot or another quantitative method should be included to support these conclusions.

Thanks for this interesting suggestion. Nevertheless, since we did not observe clear differences between groups, we did not consider useful to confirm the results by other methods.

An alternative explanation for the similar levels of pSMAD observed in patient and control samples (Figure 5) could be differences in BGN expression, which should be assessed and discussed.

R Expression levels were similar in patients’ and control cells. This is now mentioned in Results (lines 181 and 190). This was not unexpected, as patients carried a missense variant.

Round 2

Reviewer 2 Report

Comments and Suggestions for Authors
  1. Line 188: “Figure 4B” should be revised to “Figure 4B1” as the conclusion is derived from the transfection experiment done in MG63 in Figure 4B1.
  2. Line 190: “Figure 4B1-2” should be revised to “Figure 4B2-3” as the conclusion is derived from the TOPFlash reporter assay done in HEK293T in Figure 4B2-3.

Author Response

  1. ine 188: “Figure 4B” should be revised to “Figure 4B1” as the conclusion is derived from the transfection experiment done in MG63 in Figure 4B1.
  2. Line 190: “Figure 4B1-2” should be revised to “Figure 4B2-3” as the conclusion is derived from the TOPFlash reporter assay done in HEK293T in Figure 4B2-3.

The relevant figure panels are  now indicated as suggested